# Preparation of Immobilised 17β-Estradiol-Imprinted Nanoparticles onto Bacterial Cellulose Nanofibres to Use for the Removal of 17β-Estradiol from Wastewater

**DOI:** 10.3390/polym15051201

**Published:** 2023-02-27

**Authors:** İlker Koç, Koray Şarkaya, Deniz Türkmen, Süleyman Aşır, Adil Denizli

**Affiliations:** 1Department of Chemistry, Faculty of Science, Hacettepe University, Ankara 06800, Turkey; 2Department of Chemistry, Faculty of Science, Pamukkale University, Denizli 20017, Turkey; 3Department of Biomedical Engineering, Near East University, Mersin 10, Nicosia 99138, Turkey

**Keywords:** 17β-estradiol, adsorption, bacterial cellulose nanofibres, molecular imprinting

## Abstract

Estradiol, a phenolic steroid oestrogen, is one of the endocrine-disrupting chemicals (EDCs) found in natural and tap waters. The detection and removal of EDCs is attracting attention daily as they negatively affect animals’ and humans’ endocrine functions and physiological conditions. Therefore, developing a fast and practical method for the selective removal of EDCs from waters is essential. In this study, we prepared 17β-estradiol (E2)-imprinted HEMA-based nanoparticles onto bacterial cellulose nanofibres (E2-NP/BC-NFs) to use for the removal of E2 from wastewater. FT-IR and NMR confirmed the structure of the functional monomer. The composite system was characterised by BET, SEM, µCT, contact angle, and swelling tests. Additionally, the non-imprinted bacterial cellulose nanofibres (NIP/BC-NFs) were prepared to compare the results of E2-NP/BC-NFs. Adsorption of E2 from aqueous solutions was performed in batch mode and investigated via several parameters for optimisation conditions. The effect of pH studies was examined in the 4.0–8.0 range using acetate and phosphate buffers and a concentration of E2 of 0.5 mg/mL. The maximum E2 adsorption amount was 254 µg/g phosphate buffer at 45 °C. The experimental data show that the Langmuir is a relevant isotherm model for E2 adsorption. Additionally, the relevant kinetic model was the pseudo-second-order kinetic model. It was observed that the adsorption process reached equilibrium in less than 20 min. The E2 adsorption decreased with the increase in salt at varying salt concentrations. The selectivity studies were performed using cholesterol and stigmasterol as competing steroids. The results show that E2 is 46.0 times more selective than cholesterol and 21.0 times more selective than stigmasterol. According to the results, the relative selectivity coefficients for E2/cholesterol and E2/stigmasterol were 8.38 and 86.6 times greater for E2-NP/BC-NFs than for E2-NP/BC-NFs, respectively. The synthesised composite systems were repeated ten times to assess the reusability of E2-NP/BC-NFs.

## 1. Introduction

Endocrine-disrupting chemicals (EDCs) are hormonally effective substances that mimic or interfere with natural endocrine system processes in animals and humans [1]. Many environmental pollutants from various water sources, mainly wastewater treatment plants, can act as EDCs even at low concentrations, affecting the normal functions of the endocrine system and causing adverse effects on natural life and humans [2]. Significant health issues in people and animals, such as neurological and developmental impairments, cancer, and reproductive abnormalities, are significantly correlated with the build-up of EDCs in the environment and cellular structures [3]. Natural and synthetic steroids are the most common EDCs [4]. 17β-Estradiol (E2) is one of the natural oestrogens with high oestrogenic activity. It is used in birth control pills, hormone treatments, and prostate and breast cancer [5].

In contrast, E2 and its metabolite are frequently detected in aquatic ecosystems and can cause severe damage [6]. Natural oestrogens, such as E2 and estrone, and synthetic oestrogens, such as 17α-ethinylestradiol, reach the environment and wastewater through human and animal urine [7]. Human and animal faeces contain natural oestrogen [8]. The concentrations of oestrogenic hormones (mainly E2, estrone, and estriol) in wastewater from traditional biological wastewater treatment plants range from a few nanograms per litre (ng/L) to a few micrograms per litre (µg/L) [7]. These micropollutants have EDC properties, and it has been determined that many species (including humans) damage the endocrine system even at the ng/L level [9]. For this reason, when the adverse conditions caused by E2 are considered, the importance of removing it from wastewater becomes evident.

Conventional processes such as coagulation and precipitation have been applied to remove EDCs. However, it has been reported that these methods are less effective on shallow molecular weight ones (100–500 Da) [10]. Catalytic degradation, photo-catalytic degradation, or bio-degradation processes depending on the chemical, physical, or biological processes have been explored to decontaminate EDCs from wastewater so far [11,12,13]. Although it has been reported that EDCs are removed from sewage due to advanced oxidation processes such as ozonation and non-thermal plasma [14,15], the uncertainty of the fate of oxidation products raises concerns [16]. The adsorption method is widely used due to environmental sensitivity, easy applicability, and high efficiency, so it is an alternative for removing EDCs from wastewater. Although some adsorbents, including chitin [17], chitosan [18], carbon nanotubes [19], and carbonaceous compounds [20] related to adsorption methods, can be applied effectively, they also have concerns. For example, activated carbon is recommended in some studies in the literature [21]. However, the fact that large amounts of activated carbon will be needed in studies that require large-scale separation is among the reasons that make this method not economically attractive [22]. While various membrane techniques such as reverse osmosis [23] and nano- or microfiltration techniques [24] are one of the most frequently applied methods for EDC removal from wastewater, trace removal efficiency is weak since the adsorbents do not contribute to specific interactions [25]. In addition, in the removal of E2 from a complex medium such as wastewater, it remains in meagre amounts due to other pollutants [26]. Therefore, new methods with improved selectivity are needed for the selective removal of E2.

Affinity carriers have specific recognition sites between target and ligand molecules, and affinity nanofibre chromatography is an effective and exciting method to separate and purify biomolecules by combining the high-efficiency nanofibres with the selectivity of chromatography materials [27]. Due to the fibre’s small diameter and the high effective surface area compared to microfibres, the adsorption capacity is thus higher [28]. Bacterial cellulose nanofibres (BC-NFs) of biological origin, with diameters ranging from 50–100 nm, have been evaluated as alternative biomaterials in chromatography studies due to biodegradability, mechanical strength, and preparation attractiveness, as well as their very favourable morphological structures [29,30,31].

Molecular imprinting is a technology that creates specific recognition sites specific to the target molecule due to covalent or non-covalent interactions between the target molecule and the template molecule in the cross-linked polymer [32]. Molecular imprinted polymers (MIPs), frequently studied in recent years, are used in many application areas where high selectivity bonding is essential. They contain specific recognition sites for the molecule of interest [33,34]. BC-NFs are a 3-dimensional reticulated porous biomaterial formed by randomly arrayed nanofibres. Since BC-NFs can be considered biomaterials, interesting studies are carried out for biomedical, chromatographic, and biotechnological purposes; in addition, BC-NFs have some advantages related to cellulose fibres such as ease of manufacture and application, superior physical and chemical properties, low cost, accessible and high purity material, and increased water holding capacity due to the large number of -OH groups. Therefore, these composite structures are also preferred in adsorption studies of biomolecules due to their high specific surface area. In addition, molecularly imprinted BC-NFs have a significant number of application areas, such as bio-separation of biological molecules, paper, filler, removal of heavy metal ions, drug delivery, packaging material, and biosensors, due to their excellent surface area, high hydrophilicity, high purity, and significant chemical and mechanical stability [35,36,37,38].

Because of high selectivity, exhibited excellent template molecules adsorption capacity, and fast binding kinetics, a new generation composite polymeric was developed to remove E2 for this study selectively. In this study, E2-imprinted HEMA-based nanoparticles onto a bacterial cellulose nanofibre (E2-NP/BC-NFs) composite system was prepared to remove E2 from wastewater. The structure of the functional monomer was confirmed by FT-IR and NMR. First, the composite system was characterised by BET, SEM, µCT, contact angle, and swelling tests. Next, the binding capacity of E2 to E2-NP/BC-NFs was investigated by selective adsorption experiments. Experiments were repeated 10 times to determine the reusability of E2-NP/BC-NFs.

## 2. Experimental

### 2.1. Chemicals

Medium components, D-glucose, yeast extract, peptone, KH_2_PO4, and K_2_HPO4, were obtained from Merck (Darmstadt, Germany). The *Acetobacter xylinum* (ATCC 10245) strain was purchased from the United States Agricultural Research Service Culture Collection (Peoria, IL, USA). In addition, E2, cholesterol, stigmasterol, ethanol, acetonitrile, methanol, and other reagents were obtained from Sigma (St. Louis, MO, USA).

### 2.2. Preparation of BC-NFs

BC-NFs were prepared by incubating *Acetobacter xylinum* (ATCC 10245) in a Hestrin–Schramm medium. Hestrin–Schramm medium contains 20.0 g/L glucose, 10.0 g/L peptones, 10 g/L yeast extract, 4 mM KH_2_PO_4_, and 6 mM K_2_HPO_4_. The pH of the medium is adjusted between 5.1 and 5.2 using 1 M HCl. Stock bacterial culture is obtained from incubating *Acetobacter xylinum* bacteria stored at 28 °C for 3 days. BC-NFs are formed due to incubating the stock culture at a ratio of 1:10 in Petri dishes in stagnant conditions for 7 days. The synthesized BC-NFs are removed from the bacteria in the nanofibres by keeping them at 70 °C for 90 min using 1 M NaOH. Then, BC-NFs (ID: 1.15 cm) are cut, washed with distilled water many times, and finally stored at room temperature.

### 2.3. Preparation of E2-Imprinted Poly(HEMA-MATrp) Nanoparticles (E2-NP)

As previously suggested in the literature, a *N*-Methacryloyl-L-tryptophan (MATrp) monomer was first synthesised [39]. Then, preparing E2-imprinted nanoparticles was as follows: the polymerisation was carried out in a binary liquid phase mixture. The first liquid phase was an aqueous solution (10 mL) of PVA (0.187 g), SDS (28.9 mg), and sodium bicarbonate (23.4 mg). The second liquid phase was an aqueous solution (200 mL) of PVA (0.1 g) and SDS (0.1 g). The monomer phase was: using prepared MATrp monomer (100 μL), HEMA (0.7 mL), and EGDMA (1.4 mL). The prepared monomer phase was added to the first liquid phase. The mixture was homogenised (T10, Ika Labortechnik, Staufen, Germany) to obtain a mini-emulsion at 25,000 rpm. Next, the template molecule (E2, 200 µmol) was added to the mini-emulsion and mixed with a magnetic stirrer for 2 h to achieve effective monomer–pattern interaction. While mixing continued, the mini-emulsion containing the template molecule was slowly added to the second liquid phase. Then, the mixture was transferred to the glass polymerisation reactor. The reactor was mechanically stirred (300 rpm) and heated at 40 °C (Radleys Carousel 6, Essex, UK). Finally, sodium bisulphite (0.115 g) and ammonium persulfate (0.126 g) were added. Polymerisation was continued at 40 °C for 24 h. The E2-imprinted particles were washed with deionised water and ethyl alcohol to remove the unreacted monomer, surfactant, and initiator. For each washing step, the solution was centrifuged at 9000 rpm for an hour (Allegra-64R, Beckman Coulter, Brea, CA, USA) and nanoparticles were separated from the washing medium. The cleaned particles were redistributed in deionised water and stored at +4 °C. Non-imprinted poly(HEMATrp) particles were prepared by the same method without a template molecule (E2) in the polymerisation medium.

### 2.4. Preparation of E2-NP/BC-NFs

To prepare E2-NP/BC-NFs, E2-imprinted poly(HEMA-MATrp) nanoparticles were chemically immobilised on the BC-NFs surface. For this part, the surface of BC nanofibres was initially modified with 3-methacryloxypropyltrimethoxysilane (3-MPS). For this purpose, BC nanofibres are added to the aqueous solution containing excess E2-NPs at room temperature, allowing them to react for 24 h. To remove unreacted E2-NPs, the composite system was washed with water and ethanol. As a result, E2-NP/BC-NFs were lyophilised, dried, and stored for use in both characterisation and adsorption studies.

### 2.5. Swelling Tests of Composite Nanofibres

Prepared composite nanofibres for this study were dried for 36 h and then weighed with an accuracy of ±0.0001. Afterwards, the swelling degrees were calculated by keeping them in water vapour, according to Equation (1).
(1)$$\mathrm{Swelling}\;\mathrm{degree}\;(\%)=\frac{W_{s}-W_{d}}{W_{d}}\times 100$$

Firstly, composite nanofibres were dried and initially weighed. The second weighing was taken, and the % water uptake ratio was determined after drying. The weights of nanofibres before and after swelling are $W_{d}$ and $W_{s}$, respectively.

### 2.6. Characterisation Studies

The structural properties of E2-imprinted composite nanofibres were investigated using the Fourier transform infrared spectroscopic method with attenuated total reflectance (FTIR-ATR). FTIR-ATR spectra of E2-NP/BC-NFs were recorded between the 4000–400 cm^−1^ range. Additionally, the size analysis of previously prepared poly(HEMATrp) particles was determined by Nano Zetasizer (NanoS, Malvern Instruments, London, UK). For analysis, 3 mL of solution was placed in the sample well of the analyser. Light scattering at a 90° incidence angle was determined. The density of deionised water and the refractive index was 0.88 mPs and 1.33, respectively. The light scattering signal was calculated as particle number/s. The measurements were repeated 3 times, and the results were examined. The surface area of the composite nanofibres was measured by multi-point analysis using the Brunauer–Emmett–Teller (BET) method (Flowsorb II 2300, Micromeritics Instrument Corp., Norcross, GA, USA). Scanning electron microscopy (SEM (QUANTA 400F Field Emission, ThermoFisher Scientific, Waltham, MA, USA)) was applied to characterise the morphology of both E2-imprinted and non-imprinted BC-NFs. Samples were attached to carbon tape for this purpose and kept under a 15 mA current for 15 s to provide Au-Pd coating. In addition, composite nanofibre samples were kept in deep freeze, then lyophilised for 36 h (−55 °C, 0.1 mbar) in a lyophiliser (Christ Alpha 1-2 LD plus, Otto Christ AG, Memmingen, Germany) before SEM analysis. The distribution of pores and flow channels of composite nanofibres were observed using a micro-computed tomography (micro-CT) instrument (Skyscan 1172, Bruker, Billerica, MA, USA). In addition, the contact angle of E2 BS composite nanofibres was also determined by the Sessile Drop method. The KRUSS DSA 100 (Hamburg, Germany) device was used with 20 repetitions to determine the contact angle, and the results’ average was calculated.

### 2.7. Adsorption of E2

Adsorption of E2 with E2-NP/BC-NFs and NIP/BC-NFs composites from aqueous solutions was studied in batch systems. The chromatographic detection of E2 was performed using a mobile phase containing water/methanol/acetonitrile (60/30/10, *v*/*v*/*v*), the linear gradient was at 0.5 mL/min flow rate by HPLC system (Ultimate-3000, Dionex, Sunnyvale, CA, USA). The column temperature was set at 25 °C, and the monitoring wavelength at 280 nm. The effect of pH, concentration, temperature, ionic strength, and time for adsorption capacity were investigated. The measurements were repeated 3 times, and the results were examined. The average of the three measurements was taken. The effect of pH experiments was studied in a range of 4.0–8.0 with acetate and phosphate buffers, with the 0.5 mg/mL concentration of E2. E2’s aqueous solutions ranging from 0.25 to 2.0 mg/mL were used to determine the E2 concentration effect on adsorption capacity. The effect of temperature on the adsorption capacity was investigated in the range of 4–55 °C. Additionally, a NaCl solution (range 0.01–0.1 M) was used to determine ionic strength, which affects adsorption capacity.

### 2.8. Selectivity

Adsorption studies were also conducted for cholesterol (MW: 386 g/mol) and stigmasterol (MW: 412.7 g/mol) to determine the selectivity of E2-NP/BC-NFs. The chromatographic detection of E2, cholesterol, and stigmasterol was performed using a mobile phase containing water/methanol/acetonitrile (60/30/10, *v*/*v*/*v*), the linear gradient was at 0.5 mL/min flow rate by HPLC system (Ultimate-3000, Dionex). The column temperature was set at 25 °C, and the monitoring wavelength at 280 nm.

### 2.9. Reusability and Reproducibility Studies

Desorption of E2 was studied with chloroform. The E2-NP/BC-NFs were immersed in a desorption medium and stirred at room temperature for 60 min. The final concentration of E2 in the desorption medium was determined using HPLC. The desorption ratio was calculated from the amount of E2 adsorbed on the particles and the final E2 concentration in the desorption medium. The adsorption–desorption cycle was repeated ten times for prepared E2-NP/BC-NFs to determine their reusability. After desorption, E2-NP/BC-NFs were washed with 50 mM NaOH solution to regenerate and sterilise.

## 3. Results and Discussion

### 3.1. Characterisation Analysis

E2-NP/BC-NFs were prepared in three stages. The first step is the bacterial incubation of the BC nanofibres and cleaning by washing. Then, E2-NPs were prepared with the removal of the template molecule. Finally, composite nanofibres with a high surface area were synthesised with E2-NPs for selective adsorption of the E2 from wastewater.

FTIR and NMR techniques were applied to determine the structural characterisation of the MATrp monomer. FTIR spectra show that the MATrp monomer has absorption bands as -NH stretch bands at 3300 and 3500 cm^−1^. Then, the bands at 3100–3000 cm^−1^ are aromatic -CH. The absorption bands at 3000–2800 cm^−1^ represent –CH groups as aliphatic. The carbonyl (-C=O) absorption band of MATrp is seen at 1715 cm^−1^. Tensile stretching bands of C=C are seen at 1600–1670 cm^−1^. The absorption bands at 1370–1250 cm^−1^ and 1320–1210 cm^−1^ represent aromatic C-N and acidic C=O stretchings, respectively. The characteristic peaks of the MATrp monomer at ^1^H-NMR spectra are as follows; (1) 8.22 (1H, s, N-H), (2) 7.54–7.09 (4H aromatics), (3) 6.98 (1H, d, amide NH J = 5.58), (4) 5.64 (1H, t, CH2), (5) 5.32 (1H, t, CH_2_), (6) 4.99 (1H, m, CH), (7) 3.38 (2H, dd, CH_2_), (8) 6.34 (1H, d, 5-ring, J = 7.6), (9) 3.71 (3H, s, OCH_3_), (10) 1.24 (3H, t, CH3), (400 MHz, DMSO-d_6_). A broad singlet peak at 8.22 ppm corresponds to the NH proton of the amide group. Four aromatic proton signals appear as a multiplet between 7.54 and 7.09 ppm. An NH proton signal of the amide group appears as a doublet at 6.98 ppm, with a coupling constant (J) of 5.58 Hz. Two methylene proton signals appear as overlapping triplets at 5.64 and 5.32 ppm. A methine proton signal appears as a broad multiplet at 4.99 ppm. Two methylene proton signals appear as a doublet of doublets at 3.38 ppm. A proton signal appears as a doublet at 6.34 ppm with a coupling constant (J) of 7.6 Hz, indicating its location in a five-membered ring. A methyl group signal appears as a singlet at 3.71 ppm. A methyl group signal appears as a triplet at 1.24 ppm. The NMR spectrum indicates that the MATrp monomer contains an amide group, an aromatic ring, and various other functional groups. The peaks are well-resolved and show distinct coupling patterns, which suggests that the compound is relatively pure and structurally well-defined. Zeta size analyses of E2-NP and non-imprinted nanoparticles are determined. The mean size of the E2-NP is 167.7 nm, and the polydispersity is 0.300. Additionally, these values were found to be 165.3 nm and 0.210, respectively, for non-imprinted nanoparticles. The BET method calculated the specific surface area of E2-NP/BC-NFs. The surface morphology of BC-NFs is more oval and smooth. The arrangement of nanofibres is random to form interconnected pore structures and three-dimensional network structures with high porosity. Thus, the NF surface can become rougher. A rougher surface causes E2-NP/BC-NFs to generate larger specific surface areas than BC-NFs. This idea follows our study. The surface area of BC-NFs, NIP/BC-NFs, and E2-NP/BC-NFs was found to be 301.5, 305.2, and 313.2 m^2^/g, respectively, in this study [36,40,41]. In addition, the particulate systems brought into the form give the BC nanofibres the functionality to be used in different application areas. BC-NFs reveal a high surface area, showing excellent potential for separation processes. With larger specific surface areas for E2-NP/BC-NFs, the biological molecules desired to be separated reach the nanofibre surface faster with rapid mass transfer. At the same time, higher adsorption capacity can be achieved. Some studies in the literature also support this idea [42,43,44,45]. Thus, BC-NFs reveal a high surface area, showing excellent potential for separation processes in different fields of biotechnology [46,47].

The optical photograph formed at the liquid–air interface BC-NFs as a result of bacterial incubation is shown in Figure 1A. In addition, the morphological structure of E2-NP/BC/NFs was also demonstrated by μCT analysis. According to both μCTs (Figure 1B), the 3-D distribution of the pores along the nanofibres has large interconnected macropores in the membrane structure of the nanofibres. These results indicate that the high surface area of the porous structure of BC-NFs is due to the interconnected nanofibre network. This unique character of BC-NFs offers significant potential for the adsorption of biological molecules, mainly due to its rapid mass transfer. Additionally, the pore diameters are sufficiently broadly apart. The composite nanofibre structure’s architecture has macropores acting as flow channels that are much larger than the E2 size. Thus, macromolecular compounds such as E2 can easily pass through the pores of the composite bacterial cellulose without causing any clogging. Therefore, the mass transfer resistance is almost negligible.

Imprinted nanoparticles close to each other in size were obtained. SEM images of both BC-NFs and E2-NP/BC/NFs are presented in Figure 2A–D, respectively. BC-NFs have three-dimensional networks. The fibre diameters’ range is estimated between 50–100 nm. Due to the porous structure of the nanofibres, the flow resistance is meagre. This feature allows BC-NFs to be quickly tested even with highly viscous fluids such as wastewater. In addition, it can be seen that E2-imprinted nanoparticles are homogeneously incorporated into the BC-NFs structure. As mentioned, BC-NFs are nanomaterials with large, connected pores, and there are nanoparticles containing specific binding sites specific to E2 on the surfaces of these composite nanofibres. Therefore, there is no mass transfer restriction in the E2 removal process.

Contact angle measurements of BC-NFs were calculated according to the Equation (2) below:(2)$$\cos\theta_{c}=\left(\frac{\gamma_{GS}-\gamma_{LS}}{\gamma_{GL}}\right)$$

$\theta_{c}$ is the contact angle, $\gamma_{GS}$ is the gas–liquid surface tension, $\gamma_{LS}$ is the liquid–solid surface tension, and $\gamma_{GL}$ is the gas–liquid surface tension. Surfaces are described as high-energy or low-energy, and water forms a thin film on high-energy surfaces. In this case, the contact angle is zero. On low-energy surfaces, the contact angle is more significant than 90°, and the surface is hydrophobic. The contact angle measurements for BC-NFs, E2-NP/BC-NFs, and NIP/BC-NFs composite nanofibres are 34.2° ± 1.2, 45.2° ± 2.4, and 42.7° ± 1.7, respectively (Figure 3A–C). BC-NFs are highly hydrophilic due to their hydroxyl groups. As explained above, it has high energy due to its hydrophilic properties, and as a result, the contact angle value (34.2° ± 1.2) was found to be relatively low. It is observed that there is an increase in the contact angle value depending on the addition of chemicals with hydrophobic characteristics to the structure. The angle (42.7° ± 1.7) increased due to the absence of E2, the template molecule in the NIP/BC-NFs structure. Due to the high hydrophobic character of the E2 molecule in the E2-NP/BC-NFs structure, an increase in the contact angle (45.2° ± 2.4) was observed. Therefore, it is possible to say that the structure is hydrophilic. In addition, it was understood that the increasing contact angle value shows that the hydrophobic E2 molecule entered into the synthesized MIP material.

### 3.2. Swelling Test Results

The elasticity of the network structure, the presence of hydrophilic groups, crosslinkers, and the material’s porosity can be shown as the factors affecting the swelling performance of BC-NFs. However, parameters such as ionic strength, temperature, and pH also provide ideas about the swelling characteristics of BC-NFs [42]. The results of the swelling degree of BC-NFs are presented in Figure 4. The swelling degree is high for BC-NFs due to their hydrophilic structure. Additionally, water uptake capacity increases with the increased surface area of bacterial cellulose. The swelling degree of bacterial cellulose decreased following polymerisation. Finally, the swelling degree is close to each other for NIP/BC-NFs and E2-NP/BC-NFs.

### 3.3. Adsorption Studies

#### 3.3.1. Effect of pH

Adsorption of E2 from aqueous solutions was studied in the pH range of 4.0–9.0. Adsorption studies were carried out by batch system using 0.5 mg/mL E2 solution and 10 mM phosphate buffer solutions. Figure 5A shows that the E2 adsorption capacity was obtained at maximum pH of 7 as 160.05 µg/g. The highest adsorption capacity of the target molecule is generally observed at the pH value at which the imprinting process takes place for molecularly imprinted systems. This phenomenon is referred to as ‘shape memory’ [43].

#### 3.3.2. Effect of pH, Concentration, and Temperature

Adsorption tests using various aqueous solutions of E2 at pH 7.0 and concentrations between 0.25–2.0 mg/mL were carried out to investigate the effect of E2 concentration on adsorption capacity. According to adsorption experiments using MIP NFs, E2 displays affinity binding as the number of molecules interacting with the imprinted regions increases. By filling in the specific gaps, it came to equilibrium at a concentration of 1.50 mg/mL E2. The highest E2 adsorption, as shown by the adsorption studies in Figure 5A, took place at pH 7.0, which shows that interactions between E2-imprinted MIP NFs are primarily based on hydrogen bonds. The functional groups of HEMA, MATrp monomers, and E2 will not form hydrogen bonds because of the deprotonation impact at high pH. Higher or lower pH values thus impact the ligand’s affinity for binding to the template molecule, reducing the selectivity of the adsorbent.

Adsorption investigations were performed in the temperature range of 4–45 °C to determine the effects of temperature on the adsorption capacity (Figure 5B). An increase in E2 adsorption was observed with increasing temperature. These findings demonstrate that incorporating E2-imprinted nanoparticles into the structure of the MATrp functional monomer results in a hydrophobic interaction between E2 and E2-NP/BC-NFs. The increased temperature enhances the adsorption capacity and accelerates the interaction kinetics between the analyte and the binding sites in hydrophobic interactions, which occur with an increase in entropy. Due to the low kinetic energy of the molecules, the hydrophobic interactions between nonpolar adsorbates and hydrophobic adsorbents may be weaker at low temperatures. As the temperature rises, the molecules’ kinetic energy also rises, which may increase the hydrophobic interactions between the adsorbent and the adsorbate. As a result, the system’s maximum adsorption capacity increases.

#### 3.3.3. Effect of Ionic Strength (IS)

The effect of ionic strength on E2 adsorption via MIP composites was investigated using a NaCl solution varying in the 0.01–0.1 M range. The ionic strength is determined by the charge number and concentration of the cations and anions that make up the salt. Figure 6 shows that adsorption capacity decreased with increasing salt concentration. Increasing salt concentration caused a decrease in the solubility of E2, thus weakening the affinity between E2 molecules and a decline followed by E2-NP/BC-NFs in the amount of adsorption. Due to weak interactions (van der Waals interaction and hydrogen bonding) between E2 and hydroxyl groups on the surface of pHEMA nanospheres, E2 adsorption is negligible. This result also shows that no observable effect was seen on pHEMA due to the lower selectivity of E2 towards pHEMA.

#### 3.3.4. Effect of Time

Adsorption studies were performed using an E2 aqueous solution with a concentration of 1.0 mg/mL at pH 7.0 throughout time intervals ranging from 0 to 60 min to examine the effects of adsorption time. In 15 min, it was seen that 91% of the E2 solutions was adsorbed, and then the adsorption capacity reached its equilibrium value almost in 30 min (Figure 7). This result shows that the mass transfer rate of the synthesised support material is reasonably sufficient.

#### 3.3.5. Selectivity Experiments

Cholesterol and stigmasterol were selected because they have similar chemical structures to E2. Adsorption studies were also performed for cholesterol (MW: 386 g/mol) and stigmasterol (MW: 412.7 g/mol) to demonstrate the selectivity of the MIP composite system. Cholesterol and stigmasterol were loaded into the aqueous E2 solution and applied to the E2-NP/BC-NFs. E2, cholesterol, and stigmasterol were added to the methanol: water (10:90 (*v*/*v*)) solution and mixed in a sonicator at room temperature for 10 min. After reaching the adsorption equilibrium, the cholesterol and stigmasterol concentrations in the remaining solution were measured in HPLC. The distribution and selectivity coefficients ($K_{d}$) of cholesterol and stigmasterol according to E2 were calculated according to the following equation (Equation (3)):(3)$$K_{d}=\left(\frac{C_{i}-C_{f}}{C_{f}}\right)\frac{v}{m}$$

For the equation (mL/g), $C_{i}\;$and $C_{f}$ are initial and final concentrations (mg/mL), *v* is the volume of aqueous solution (mL), and m is the weight of the composites (g).

The selectivity coefficient for binding a molecule in the presence of competitive species can be calculated from the equilibrium binding data (Equation (4)).
(4)$$k=\frac{K_{E2}}{K_{competitive}}$$

“*k*” represents the selectivity coefficient and *K_competitive_* means cholesterol or stigmasterol. Comparing the *k* values of the MIP composite with the *k* values of cholesterol and stigmasterol allows for an evaluation of the effect of molecular imprinting on selectivity. The relative selectivity coefficient *k*′ can be defined according to the following equation:(5)$$k^{’}=\frac{k_{imprinted}}{k_{control}}$$

Table 1 lists the $K_{d}$, *K*, and *K*′ values used in selectivity studies. The MIP-composite had a distribution coefficient ($K_{d}$) value that was higher than NIP/BC-NFs, the control group’s distribution coefficient. The $K_{d}$ value for E2 increased, whereas it tended to decrease for the competitor molecules cholesterol and stigmasterol. The relative selectivity coefficient measures how well recognition sites attach to compounds with E2 imprinting. Results indicated that the MIP-composite had relative selectivity coefficients for E2/cholesterol and E2/stigmasterol, which were 8.38 and 86.6 times greater than NIP/BC-NFs. Selectivity depends on the imprinted cavities’ form and size memory. The chemical structure of the competing molecules contains linear chains, which make these molecules more hydrophobic than E2. As a result, the competing molecules are more hydrophobic than E2.

### 3.4. Physicochemical Analysis of Adsorption

#### 3.4.1. Adsorption Isotherms

Adsorption isotherm models explain the relationships between the amount of solute adsorbed on any adsorbent ($Q_{e}$) and the dissolved substance ($C_{e}$) concentration in the liquid phase at a given temperature. Langmuir and Freundlich adsorption isotherm models were applied in this study. The Langmuir adsorption isotherm model; suggests that adsorption is controlled by homogeneous, monolayer, and similar binding sites. It also states that the energy distribution does not change during adsorption, and there are no interactions between adjacent adsorption sites. The Langmuir adsorption model is defined according to the equation (Equation (6)) given below:(6)$$\frac{C_{eq}}{Q}=\left[\frac{1}{Q_{\max}Xb}\right]+\left[\frac{C_{eq}}{Q_{\max}}\right]$$

In these equations, *Q* is the adsorbed E2 amount (µg/g); $C_{eq}$, equilibrium E2 concentration (mg/mL) in solution; $Q_{\max}$, maximum E2 adsorption capacity (µg/g); and *b*, the Langmuir constant (mL/mg). The equilibrium and maximum are referred to as *eq* and max, respectively.

The Freundlich adsorption model argues that the adsorption is heterogeneous and multi-layered on the adsorbent surface and is defined according to the equation (Equation (7)) given below:(7)$$\ln Q_{eq}=lnK_{f}+\frac{1}{n}C_{eq}$$

According to the Freundlich isotherm model, $Q_{eq}\;$is adsorption amount, $K_{f}$ is the Freundlich constant, “*n*” stands for Freundlich heterogeneity index. The $K_{f}$ and 1/*n* values can be obtained by plotting the linear graph of ln$Q_{E}$ vs. ln$C_{e}.$

The Langmuir and Freundlich isotherm models were applied for the investigation of adsorption of E2 onto BC-NFs in this study. According to this, $K_{f}$ and 1/*n* values indicate the surface heterogeneity of the system in the Freundlich isotherm model. The closer the 1/*n* value is to 1, the more homogeneity of the system; the closer the “*n*” value is to zero, the more significant the heterogeneity of the system. Therefore, considering these results and the correlation coefficients ($R^{2}$), the experimental data show that they are compatible with the Langmuir isotherm model ($R^{2}$: 0.97). E2 indicates that the binding sites on the MIP composite surface are co-energy and have the lowest lateral interaction. The results are also given in Table 2.

#### 3.4.2. Adsorption Kinetics

Mathematical approaches for adsorption processes are applied to explain mass transfer and chemical interaction mechanisms. It is also possible to obtain information about adsorption kinetic studies and the steps affecting the adsorption rate. Pseudo-first- and second-order kinetic models were applied to determine the adsorption kinetics of E2.

For pseudo-first-order:(8)$$\log(Q_{e}-Q)=LogQ_{e}-k_{1}\frac{t}{203}$$

In this equation, $Q_{e}$ and *Q* (µg/g) are the adsorption capacities at equilibrium and at time “*t*”, respectively; $k_{1}$ (min^−1^) represents the pseudo-first order adsorption rate constant.

For pseudo-second-order:(9)$$\frac{t}{Q}=\frac{1}{k_{2}Q^{2}{}_{e}}+\frac{1}{Q_{e}}$$

In these equations, $k_{2}$ (min^−1^) represents the pseudo-second-order adsorption rate constant.

Table 3 provides an overview of the pseudo-first- and second-order kinetic constants of E2-NP/BC-NFs. When the correlation coefficients and *Q_e_* values are examined, the adsorption studies with E2-NP/BC-NFs fit the pseudo-second-order kinetic model. These findings demonstrate the chemical modulation of the adsorption kinetics and adsorption kinetics are chemically controlled.

#### 3.4.3. Thermodynamic Analysis

The adsorption process occurs with the attraction force formed between E2 and its specially designed binding sites. As a result of the interaction of E2 with its specific binding sites, an increase in entropy (*S*) is observed. Thermodynamic parameters allow us to obtain important information about energy changes during adsorption. Gibbs free energy value change (Δ*G*) will enable us to determine whether the adsorption process is voluntary or involuntary. All these thermodynamic energy change values (Δ*H*, Δ*S*, Δ*G*) can be calculated using the equation below.
(10)$$\ln K_{L}=\frac{-\Delta H{}^{\circ}}{RT}+\frac{\Delta S{}^{\circ}}{R}$$

In this equation, $K_{L}$ is the Langmuir adsorption coefficient; *R*: (8.3145 J/(mol K)) is the ideal gas constant; “*T*” (K) denotes temperature.

Table 4 presents the thermodynamic parameters of MIP NFs. The adsorption process occurs with the attraction force formed between E2 and its specially designed binding sites. As a result of the interaction of E2 with its specific binding sites, an increase in entropy (*S*) is observed.

### 3.5. Reusability

The synthesised composite systems were repeated ten times with 1 mg/mL of E2 aqueous solution at pH 7.0 to assess the reusability of E2-NP/BC-NFs. The findings show that throughout ten cycles, no noticeable decline in adsorption capacity was found (Figure 8). In the first cycle, the desorption rate was calculated to be 95%, while in the tenth cycle, it was determined to be 90%. These findings demonstrate the reusability of E2-NP/BC-NFs in adsorption investigations.

### 3.6. Comparison with E2-Imprinted Adsorbents

In the literature, studies have reported on the selective removal of EDCs (Table 5). As different adsorbents have been proposed for the adsorption of E2, the importance of studies using molecularly imprinted polymers is emerging. Farber et al. prepared MIPs for E2 removal using covalent and non-covalent interactions. For this purpose, 4-vinyl benzene as a covalent monomer, methacrylic acid as a non-covalent monomer, ethylene glycol dimethacrylate as crosslinker, and acetonitrile was used as porogen. It was concluded that the polymers obtained have the highest adsorption capacity (10.73 μg/g) due to the imprinting process using covalent interactions [44]. Denizli et al., obtained polymers with the highest adsorption capacity (10.73 μg/g) due to the covalent interactions’ imprinting process. E2-imprinted particle-embedded poly(HEMA) cryogels were prepared for E2 removal. The composite cryogels achieved an adsorption capacity of 5.32 mg/g. As a result of the adsorption studies performed in the presence of cholesterol and stigmasterol selected as competitive molecules, it was concluded that there is 7.6 times more selective binding to cholesterol and 85.8 times more to stigmasterol [45]. In the study using lake water as a natural source, Meng et al. prepared E2-imprinted microspheres using the non-covalent imprinting technique. The maximum adsorption capacity for E2 reached 380 nmol/g using MIP microspheres. Furthermore, the imprinting factor was calculated as 35 [46]. E2-imprinted methacrylate-based microparticles were prepared by Lai et al. As a result of adsorption studies, it was reported that high recovery rates of up to 97% were obtained, and the adsorption capacity was determined as 15 mg/g [47]. Xiong et al. reported that switchable zipper-like thermoresponsive molecularly imprinted polymers reach a higher adsorption capacity (8.78 mg/g) and more substantial selectivity (imprinting factor was 3.18) [48]. Gao et al. prepared magnetic MIPs using nanoparticles of Fe_3_O_4_ with acrylamide as a functional monomer to separate and determine E2 from milk. It was reported that the adsorption performance of the MIP system was 12.62 mg/g [49]. He et al. reported using core-shell molecularly imprinted polymers fixed to the SiO_2_ surface to separate and purify E2 from marine sediment for the first time. Studies using 3-aminopropyltrimetyoxysilane as a coupling agent with methacrylic acid as a functional monomer were performed with the MIP solid-phase extraction method with HPLC. This study’s result determined that the adsorption capacity for the MIP system was 468.3 µg/g, while it was 146.7 µg/g for NIP [50]. Xia et al. used surface-modified magnetic nanoparticles as core-shell MIPs by ultrasonication-assisted synthesis to selectively remove E2 from aqueous media. The results show that E2-imprinted MIPs achieved a good recovery of 95.8% [51]. Peng et al. prepared photonic magnetic responsive molecularly imprinted microspheres via seed polymerisation for the removal of E2. It was reported that MIP microspheres show good adsorption characteristics for E2 with an adsorption capacity of 0.84 mg/g, with fast binding kinetics (*K_d_* = 26.08 mg L^−1^) [52]. In a study using MIPs created by surface imprinting of magnetic graphene oxide, it was reported that E2 was removed quickly and selectively from acetonitrile solutions of E2. Adsorption studies show that the kinetic value and binding capacity of the MIP system were 0.0062 g (mg/min) and 4.378 μmol/g, respectively [53]. Our study achieved the maximum E2 adsorption amount of 254 µg/g at 45 °C using E2-NP/BC-NFs. Therefore, these results show that E2-NP/BC-NFs have very promising adsorbents, showing comparable performance in the selective and effective removal of E2 as useful adsorbents, which is highly attractive in the literature.

## 4. Conclusions

HEMA-based nanoparticles imprinted with E2 on BC-NFs were used as a novel adsorbent to selectively remove E2 from aqueous solutions. The prepared composite nanofibres obtained adsorbents with high fast binding capacity and high selectivity. The maximum amount of E2 adsorption at pH 7 was 160.05 µg/g. Experimental data show that the Langmuir isotherm model applies to E2 adsorption. The related kinetic model was also the so-called quadratic kinetic model. Cholesterol and stigmasterol were used as competing steroids in selectivity studies. Composite systems were synthesised ten times to test the reusability of E2-NP/BC-NFs. Finally, no noticeable decrease in adsorption capacity was reached. E2 has a low water solubility, making it difficult to remove from aqueous solutions such as wastewater. The newly synthesised substance’s capacity to efficiently catch and remove the hormone will depend on its solubility characteristics and how it interacts with the aqueous solution when used to remove E2 from wastewater. Due to E2’s limited solubility, it may have a tendency to adsorb to solid objects or wastewater particles rather than staying in the solution. This can make it challenging to remove using standard wastewater treatment techniques. The newly created substance may be used to boost the adsorption of E2 onto its surface, enhancing the removal of the EDC from wastewater. Affinity nanofibres gain importance in separation processes by combining the high efficiency of nanofibres with an affinity for a particular molecule and high selectivity. These results show that composite nanofibres offer an important long-lasting alternative for removing EDCs.

## Figures and Tables

**Figure 1 polymers-15-01201-f001:**
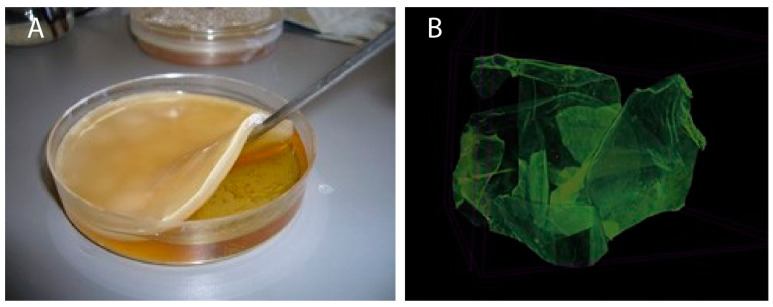
Optical and μCT images. (**A**) Optical photograph of BC nanofibres; (**B**) μCT image of BC nanofibres.

**Figure 2 polymers-15-01201-f002:**
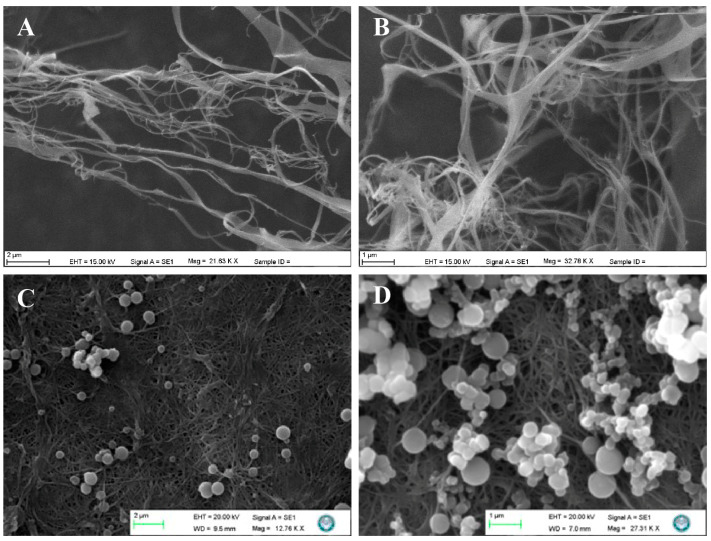
SEM images. (**A**,**B**) BC-NFs; (**C**,**D**) E2-NP/BC-NFs.

**Figure 3 polymers-15-01201-f003:**
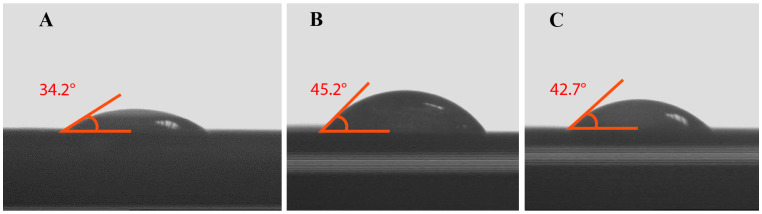
Contact angle images. (**A**) BC-NFs; (**B**) E2-NP/BC-NFs; (**C**) NIP/BC-NFs.

**Figure 4 polymers-15-01201-f004:**
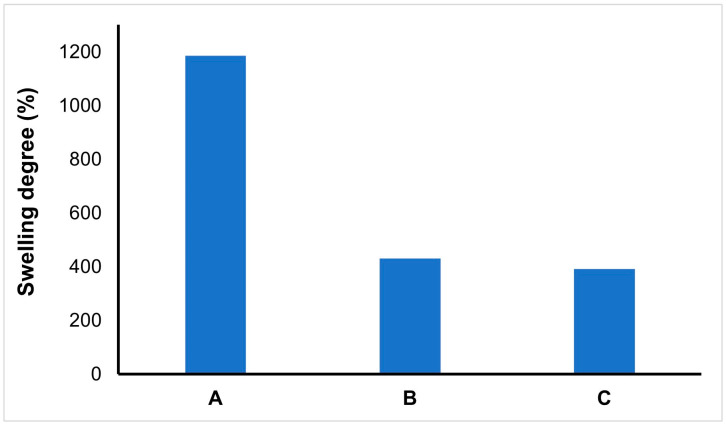
% Swelling degree. (**A**) BC-NFs; (**B**) NIP/BC-NFs; (**C**) E2-NP/BC-NFs.

**Figure 5 polymers-15-01201-f005:**
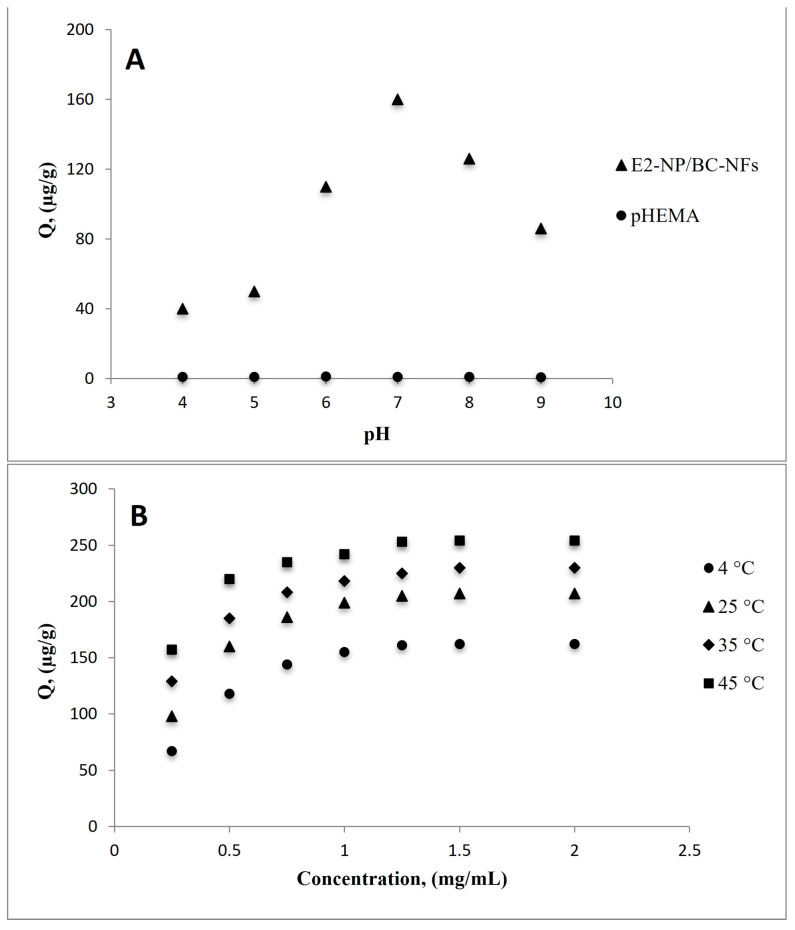
(**A**) The effect of pH on adsorption capacity: ($c_{0}$: 0.5 mg/mL, IS: 0.0, $W_{particles}$: 100 mg, t: 25 °C); (**B**) The effect of E2 concentration with different temperatures on adsorption capacity: ($c_{0}$: 0.5–2.0 mg/mL, pH: 7.0, IS: 0.0, $W_{particles}$: 100 mg, t: 25 °C).

**Figure 6 polymers-15-01201-f006:**
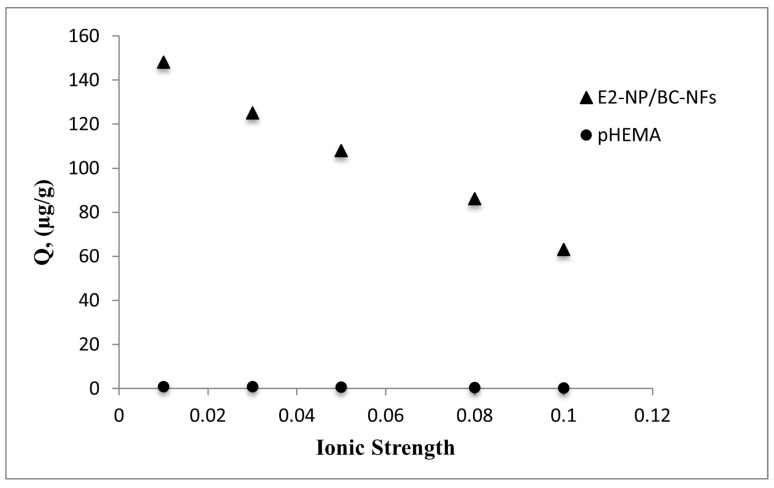
The effect of ionic strength on adsorption capacity: ($c_{0}$: 0.5 mg/mL, pH: 7.0, IS (0.01–0.1), $W_{particles}$: 100 mg, t: 25 °C).

**Figure 7 polymers-15-01201-f007:**
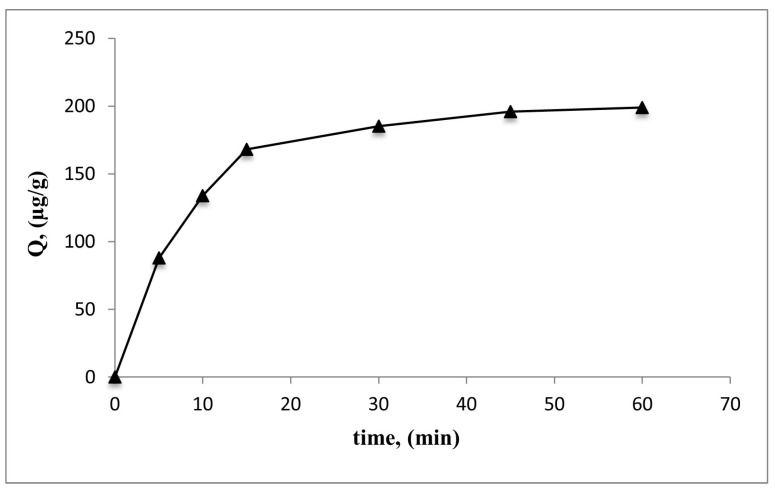
The effect of E2 time on adsorption capacity: ($c_{0}$: 1.0 mg/mL, pH: 7.0, IS: 0, $W_{particles}$: 100 mg, t: 25 °C).

**Figure 8 polymers-15-01201-f008:**
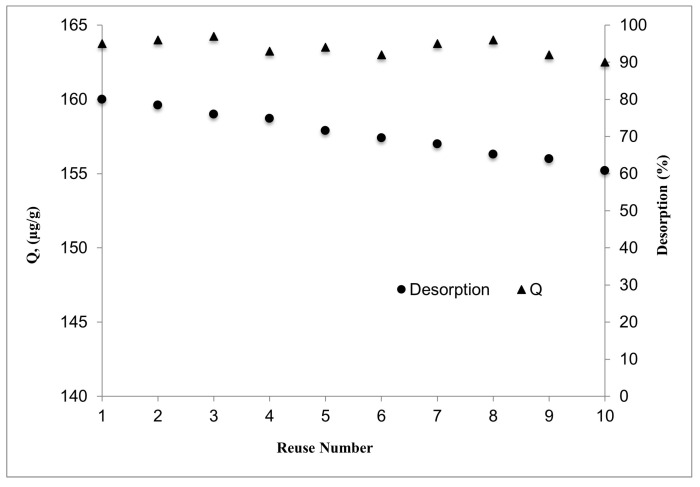
The reusability of E2-NP/BC-NFs: ($c_{0}$: 1.0 mg/mL, pH:7, I.S. 0, $W_{particles}$: 100 mg, t: 25 °C).

**Table 1 polymers-15-01201-t001:** Selectivity coefficients (*k*); relative selectivity coefficients (*k*′) and values of E2-NP/BC-NFs and NIP/BC-NFs.

Compound	NIP/BC-NFs		E2-NP/BC-NFs		
	$K_{d}$	*k*	$K_{d}$	*k*	*k*′
E2	87.1		846.0		
Cholesterol	39.7	2.19	46.0	18.36	8.38
Stigmasterol	188.4	0.465	21.0	40.28	86.6

**Table 2 polymers-15-01201-t002:** Parameters of Langmuir and Freundlich isotherms.

Langmuir Isotherm Model	Freundlich Isotherm Model
*Q*_max_: 277.80 µg/g	*K_F_*: 184.88
*K_L_*: 2.25 mL/µg	*n*: 2.82
*R*^2^: 0.96	*R*^2^: 0.84

**Table 3 polymers-15-01201-t003:** Pseudo-first- and second-order kinetic constants of E2-NP/BC-NFs.

	Kinetic Parameters	
Pseudo-first-order	*k*_1_ (×10^2^ min^−1^)	0.086
	*Q_e_*, cal (µg/g)	150.21
	*R* ^2^	0.982
Pseudo-second-order	*k*_2_ (×10^3^ g µg^−1^ min^−1^)	0.00071
	*Q_e_*, cal (µg/g)	222.22
	*R* ^2^	0.998

**Table 4 polymers-15-01201-t004:** Thermodynamic parameters of E2-NP/BC-NFs.

Temperature, °C	Δ*G*° (kJ/mol)	Δ*H*° (kJ/mol)	Δ*S*° (kJ/mol)
4	−12.55		
25	−14.33		
35	−15.18		
45	−16.03	−10.93	0.084

**Table 5 polymers-15-01201-t005:** Comparison of removal capacity of E2 with different E2-imprinted adsorbents.

Method/Functional Monomer/Polymer	% Removal	Sample	Ref.
Covalent MIP/4-vinyl benzene-methacrylic acid/polymer	10.73 μg/mg	Aqueous media	[44]
MIP particle-embedded poly(HEMA) cryogel	5.32 mg/g	Aqueous media	[45]
Noncovalent MIP/Acrylamide-trimethylpropanol trimethacrylate/microsphere	380 nmol/mg	Aqueous media	[46]
MIP/methacrylic acid-ethylene glycol dimethcarylate/submicron particles	15 mg/g	Aqueous media	[47]
stimuli-responsive MIP/Acrylamide-2-2-acrylamide-2-methyl propane sulfonic acid/polymer	8.78 mg/g	Acetonitrile solutions	[48]
Surface MIP/ Fe_3_O_4_@Acrylamide/nanoparticle	12.62 mg/g	Milk	[49]
Core-shell MIP/ 3-aminopropyltrimetyoxysilane-methacrylic acid/polymer	468.3 µg/g	Marine sediment	[50]
Core-shell MIP/methacrylic acid-ethylene glycol dimethcarylate/magnetic nanoparticle	>95	Aqueous media	[51]
Photonic-magnetic responsive MIP/ Fe_3_O_4_@SiO_2_-KH_570_-4-[(4-methacryloyloxy) phenylazo]benzoic acid/nanoparticle	315.6 ± 10.1 μg/g	Milk	[52]
Nanosized substrate MIP-GO-Fe_3_O_4_ /nanoparticle	4.378 µmol/g	Acetonitrile solutions	[53]
Nanoparticle MIP/HEMA-MATrp/bacterial cellulose nanofibre	254 µg/g	Aqueous media	Our study

## Data Availability

The authors can confirm that all relevant data are included in the article.

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
