# Peer review of "Preparation of Immobilised 17β-Estradiol-Imprinted Nanoparticles onto Bacterial Cellulose Nanofibres to Use for the Removal of 17β-Estradiol from Wastewater"

_polymers, 2023, doi:10.3390/polym15051201_

Round 1

Reviewer 1 Report

The 17β-estradiol (E2)-imprinted HEMA-based nanoparticles onto bacterial cellulose nanofibers (E2-NP/BC-NFs) were prepared and used for the removal of E2 from wastewater. Some interesting results were achieved; however, the specific comments should be considered.

1. Page 2 “Molecularly imprinted BC-NFs was used for separation of biological molecules, removal of heavy metal ions and drug delivery studies [35–38].” It was suggested to describe more detailly. The innovation of this study should be highlighted.

2. Page 3 “17-β estradiol (E2) imprinted poly(HEMA-MATrp) nanoparticles were immobilized on BC-NF surface.” Is there an interaction between E2-NP and BC-NFs? Whether the ratio of the two materials change will affect the absorption capacity?

3. “The fiber diameters range estimated between 50-100 nm.” But it can not be observed clearly in Fig.2. The scale of the SEM images was too large and it was suggested to re-label.

4. What causes the difference of the contact angle of three materials? It was suggested to analyze the reason rather than describing the picture. Other discussions including the influencing factors on adsorption studies also exist the problem.

5. Page 14 “3.6. Comparison with E2-imprinted adsorbents” It was suggested to list the comparison of these adsorbents intuitively. The terms such as the raw materials, preparation methods, adsorption capacity and reusability can be compared.

6. The difference of the E2-NP/BC-NFs and other E2 molecularly imprinted absorbents should be added in introduction and discussion.

Author Response

Reviewer 1

  1. Page 2 “Molecularly imprinted BC-NFs was used for separation of biological molecules, removal of heavy metal ions and drug delivery studies [35–38].” It was suggested to describe more detailly. The innovation of this study should be highlighted.

The additional explanations were added in the manuscript as given below:

In addition, Molecularly imprinted BC-NFs have a significant amount of application areas such as bio-separation of biological molecules, paper, filler, removal of heavy metal ions, drug delivery, packaging material, and biosensors due to their excellent surface area, high hydrophilicity, high purity, and significant chemical and mechanical stability.

Because of high selectivity, exhibited excellent template molecules adsorption capacity and fast binding kinetics, a new generation composite polymeric was developed.

  1. Page 3 “17-β estradiol (E2) imprinted poly(HEMA-MATrp) nanoparticles were immobilized on BC-NF surface.” Is there an interaction between E2-NP and BC-NFs? Whether the ratio of the two materials change will affect the absorption capacity?

The immobilization process was performed via chemical modification. It is described in the section of ‘’ Preparation of E2-NP/BC-NFs ‘’ as ‘’ To prepare E2-NP/BC-NFs, E2 imprinted poly(HEMA-MATrp) nanoparticles were chemically immobilized on the BC-NFs surface. For this part, the surface of BC nanofibers was initially modified with 3-methacryloxypropyltrimethoxysilane (3-MPS). For this aim, BC nanofibers are added to the aqueous solution containing an excess amount of E2-NPs at room temperature, allowing them to react for 24 hours. To remove unreacted E2-NPs, the composite system was washed with water and ethanol. As a result, E2-NP/BC-NFs were lyophilized, dried, and stored for use in both characterization and adsorption studies. ‘’

  1. “The fiber diameters range estimated between 50-100 nm.” But it can not be observed clearly in Fig.2. The scale of the SEM images was too large and it was suggested to re-label.

Figures 2 A, B, C, and D were updated.

A.)

B)

C)

D)

  1. What causes the difference of the contact angle of three materials? It was suggested to analyze the reason rather than describing the picture. Other discussions including the influencing factors on adsorption studies also exist the problem.

Thank you for your comments. The relevant section was revised as given below.

ɵ c is the contact angle, ɣ_GS is the gas-liquid surface tension, ɣ_LS is the liquid-solid surface tension, and ɣ_GL is the gas-liquid surface tension. Surfaces are described as high-energy or low-energy, and water forms a thin film on high-energy surfaces. In this case, the contact angle is zero. On low-energy surfaces, the contact angle is more significant than 90º, and the surface is hydrophobic.  The contact angle measurements for BC-NFs, E2-NP/BC-NFs, and NIP-BC-NFs composite nanofibers are 34.2°±1.2, 45.2°±2.4 and 42.7°±1.7, respectively (Figure 3. A-C). BC-NFs are highly hydrophilic due to their hydroxyl groups. As explained above, it has high energy due to its hydrophilic properties, and as a result, the contact angle value (34.2°±1.2) was found to be relatively low. It is observed that there is an increase in the contact angle value depending on adding chemicals with hydrophobic character to the structure. The angle (42.7°±1.7) increased due to the absence of E2, the template molecule in the NIP BC-NFs structure. Due to the high hydrophobic character of the E2 molecule in the E2-NP/BC-NFs structure, an increase in the contact angle (45.2°±2.4) was observed. So, it is possible to say that the structure is hydrophilic. In addition, it was understood that the increasing contact angle value shows the hydrophobic E2 molecule entered into the synthesized MIP material.

  1. Page 14 “3.6. Comparison with E2-imprinted adsorbents” It was suggested to list the comparison of these adsorbents intuitively. The terms such as the raw materials, preparation methods, adsorption capacity and reusability can be compared.

Thank you very much for you comments. Comparison of E2-imprinted adsorbents part was revised as given below.

In the literature, studies are reported on the selective removal of EDCs. As different adsorbents have been proposed for the adsorption of E2, the importance of studies using molecularly imprinted polymers is emerging. Farber et al. prepared MIPs for 17β-estradiol removal using covalent and non-covalent interactions. For this purpose, 4-vinyl benzene as a covalent monomer, besides methacrylic acid as a non-covalent monomer, ethylene glycol dimethacrylate crosslinker, and acetonitrile were used as porogen. It was concluded that the polymers obtained have the highest adsorption capacity (10.73 μg/g) due to the imprinting process using covalent interactions [44]. Denizli et al., the obtained polymers have the highest adsorption capacity (10.73 μg/g) due to the covalent interactions' imprinting process. E2 imprinted particle-embedded poly(HEMA) cryogels prepared for 17β-Estradiol (E2) removal. The composite cryogels achieved an adsorption capacity of 5.32 mg/g. As a result of the adsorption studies performed in the presence of cholesterol and stigmasterol selected as competitive molecules, it was concluded that there is 7.6 times more selective binding to cholesterol and 85.8 times more to stigmasterol.[45]. In the study using lake water as a natural source, Meng et al. prepared estradiol-imprinted microspheres using the non-covalent imprinting technique. The maximum adsorption capacity for estradiol reached 380 nmol/g using MIP microspheres. Furthermore the imprinting factor was calculated as 35 [46]. Lai et al., E2 imprinted methacrylate-based microparticles were prepared by Lai et al.. As a result of adsorption studies, it was reported that high recovery rates of up to 97% were obtained, and the adsorption capacity was determined as 15 mg/g [47]. Xiong et al. reported that switchable zipper-like thermoresponsive molecularly imprinted polymers reach higher adsorption capacity (8.78 mg/g) and more substantial selectivity (imprinting factor was 3.18) [48]. Gao et al. prepared magnetic MIPs using nanoparticles of Fe3O4 with acrylamide as a functional monomer to separate and determine E2 from milk. It was reported that the adsorption performance of the MIP system was 12.62 mg/g [49]. He et al. reported using core-shell molecularly imprinted polymers fixed to the SiO2 surface to separate and purify β-estradiol from marine sediment for the first time. Studies using 3-aminopropyltrimetyoxysilane as a coupling agent with methacrylic acid as a functional monomer were performed with MIP solid-phase extraction method with HPLC.  This study's result determined that the adsorption capacity for the MIP system was 468.3 µg/g, while it was 146.7 µg/g for NIP [50]. Xia et al. used surface-modified magnetic nanoparticles as core-shell MIPs by ultrasonication-assisted synthesis to selectively remove E2 from aqueous media. The results show that  E2-imprinted MIPs achieved a good recovery of 95.8% [51]. Peng et al. prepared photonic-magnetic responsive molecularly imprinted microspheres via seed polymerization to the removal of E2. It was reported that MIP-microspheres show good adsorption characteristics for E2 with an adsorption capacity of 0.84 mg/g) with fast binding kinetics (Kd = 26.08 mg L−1) [52]. In a study using MIP created by surface imprinting of magnetic graphene oxide, it was reported that E2 was removed quickly and selectively from acetonitrile solutions of E2. Adsorption studies show that the kinetic value and binding capacity of MIP system were 0.0062 g (mg/min) and 4.378 μmol/g, respectively [53]. Our study achieved the maximum E2 adsorption amount of 254 µg/g at 45°C using E2-Imprinted Nano-Particles onto Bacterial Cellulose Nanofibers. So, these results show that E2-NP/BC-NFs have very promising adsorbents showing comparable performance in the selective and effective removal of E2 as useful adsorbents, which is highly attractive in the literature.

Table 4. Comparison of removal capacity of 17β-Estradiol with different E2-imprinted adsorbents.

Method/Functional monomer/Polymer

% Removal

Sample

Ref

Covalent MIP/ 4-vinyl benzene-methacrylic acid /polymer

10.73 μg/mg

Aqueous media

[44]

MIP particle-embedded poly(HEMA) cryogel

5.32 mg/g

Aqueous media

[45]

Non covalent MIP/ Acrylamide- trimethylpropanol trimethacrylate/microsphere

380 nmol/mg

Aqueous media

[46]

MIP/ methacrylic acid-ethylene glycol dimethcarylate/ submicron particles

15 mg/g

Aqueous media

[47]

stimuli-responsive MIP/Acrylamide-2- 2-acrylamide-2-methyl propane sulfonic acid/polymer

8.78 mg/g

Acetonitrile solutions

[48]

Surface MIP/ Fe3O4@Acrylamide/nanoparticle

12.62 mg/g

Milk

[49]

Core-shell MIP/ 3-aminopropyltrimetyoxysilane-methacrylic acid/polymer

468.3 µg/g

Marine sediment

[50]

Core-shell MIP/ methacrylic acid-ethylene glycol dimethcarylate /magnetic nanoparticle

>95

Aqueous media

[51]

Photonic-magnetic responsive MIP/ Fe3O4@SiO2-KH570-4-[(4-methacryloyloxy) phenylazo]benzoic

acid /nanoparticle

315.6 ± 10.1 μg/g

Milk

[52]

Nanosized substrate MIP /Acrylamide- -GO-Fe3O4 /nanoparticle

4.378 µmol/g

Acetonitrile solutions

[53]

Nanoparticle MIP/HEMA-MATrp/bacterial cellulose nanofiber

254 µg/g

Aqueous media

Our study

  1. The difference of the E2-NP/BC-NFs and other E2 molecularly imprinted absorbents should be added in introduction and discussion.

Thank you for your advice; we updated of introduction part with the information from the literature, including why the use of molecularly imprinted bacterial cellulose nanofiber (MIP NFs) was preferred in this study. The updated part is below:

The results of E2-NP/BC-NFs and other E2 molecularly imprinted absorbents were added and discussed comprehensively in section title 3.6. Comparison with E2-imprinted adsorbents.

Molecular imprinting is a technology that creates specific recognition sites specific to the target molecule due to covalent or non-covalent interactions between the target molecule and the template molecule in the cross-linked polymer [32]. Molecular imprinted polymers (MIPs), frequently studied in recent years, are used in many application areas where high selectivity bonding is essential. They contain specific recognition sites for the molecule of interest [33,34]. BC-NFs are a 3-dimensional reticulated porous biomaterial formed by randomly arrayed nanofibers. Since BC-NFs can be considered biomaterials, interesting studies are carried out for biomedical, chromatographic, and biotechnological purposes; in addition, BC-NFs have some advantages related to cellulose fibers such as ease of manufacture and application, superior physical and chemical properties, low cost, accessible and high purity material, and increased water holding capacity due to the large number of -OH groups. Therefore, these composite structures are also preferred in adsorption studies of biomolecules due to their high specific surface area. In addition, Molecularly imprinted BC-NFs have a significant amount of application areas such as bio-separation of biological molecules, paper, filler, removal of heavy metal ions, drug delivery, packaging material, and biosensors due to their excellent surface area, high hydrophilicity, high purity, and significant chemical and mechanical stability [35–38].

Because of high selectivity, exhibited excellent template molecules adsorption capacity and fast binding kinetics, a new generation composite polymeric was developed to remove E2 for this study selectively. In this study, E2-imprinted HEMA-based nanoparticles onto bacterial cellulose nanofibers (E2-NP/BC-NFs) composite system was prepared to remove E2 from wastewater. Structure of functional monomer was confirmed by FT-IR and NMR. First, the composite system was characterized by BET, SEM, µCT, contact angle, and swelling tests. Next, the binding capacity of E2 to E2-NP/BC-NFs was investigated by selective adsorption experiments. Experiments were repeated 10 times to determine the reusability of E2-NP/BC-NFs.

Reviewer 2 Report

Recommendation: Major revision

 This is a review of the manuscript entitled: ‘Preparation of Immobilized 17β-Estradiol Imprinted NanoParticles onto Bacterial Cellulose Nanofibers to Use for the Removal of Estradiol from Wastewaters’. This is an interesting manuscript exploring the use of modified 17β-estradiol (E2)-imprinted HEMA-based nanoparticles onto bacterial cellulose nanofibers (E2-NP/BC-NFs) for adsorption of E2 from wastewater. Overall, the research writing is pretty decent making it easily understandable for the audience. This work would be more interesting to Polymers Journal if some revision have been made.

To me the author present many results but lack in some of the discussion part on each experiment, and the results still required connectivity between experiments. I also suggest the authors state more of their novelty of this contribution. The low solubility of E2 should also be more discuss as it can be interfere/support the adsorption performance.

1.         The abstract had merely results. I recommend putting more of the new finding (with discussion) in the abstract. This would definitely make this article stand out.

2.         The introduction could have been more attractive in term of stating the new finding and stated while this quite complicated material would be more interesting than other known effective methods.

3.         The preparation of 17β-Estradiol-Imprinted poly(HEMA-MATrp) nanoparticles (E2-NP) is quite complicated. Can the authors elaborate how their material reproductivity is reliable? Are there specific tool to confirm it? It should be mentioned in the methodology section.

4.         Chemical vendor & chemical analyses are needed. It may not need to be in the manuscript, but it could have been in the supporting information. How did the authors actually measure E2 concentration with “its absorbance at 280 nm”? I assume the author used SPE for the E2 measurement? Please explain.

5.         What is the purpose of the experiment in Section 2.8. If the author need to prove the material reusability, why does the author need to change the contaminant and why does the author need to use the sonicator? Wouldn’t it interfere the interpretation of the adsorption studies? Please explain.

6.         What is the main purpose of determining the contact angle? The authors should have given more explanation of the contact angle. It shouldn’t be for confirming the material hydrophilicity, should it?

7.         E2 is known to have a very low water solubility. The detected amount is normally in a nanogram or microgram level. Please elaborate how this would affect this newly synthesized material? Can the contact angle help us explaining anything?

8.         There was insufficient discussion for the adsorption studies, especially effect of pH. Please provide more discussion.  

9.         The effect of ionic strength was different between MIP and pHEMA. Why is that? Please explain.

10.      Please discuss and compare (with other findings) more on the maximum adsorption capacity of the material.

11.      I am skeptical to accept that the BET slight difference was from “the cavity formed by the imprinting of the estradiol molecule”. Please consider rewrite.

12.      The NMR result has no discussion. Please provide more.

13.      There are several inconsistency in the use of abbreviation, which makes it quite difficult to follow. Please consider reorganize it.

14.      At the end, however, I'd like the authors to reflect on the value of manuscript by telling readers how all this research material could be used in practice and applied to the conventional industrial wastewater treatment plant. Without some application, the value of this work will be lost.

Author Response

Reviewer 2

  1. The abstract had merely results. I recommend putting more of the new finding (with discussion) in the abstract. This would definitely make this article stand out.

The abstract part was revised as given below:

Estradiol, a phenolic steroid estrogen, is one of the endocrine-disrupting chemicals (EDCs) found in natural and tap waters. The detection and removal of EDCs are attracting attention daily as it negatively affects animals’ and humans' endocrine functions and physiological conditions. Therefore, developing a fast and practical method for the selective removal of EDCs from waters is essential. In this study, we prepared 17β-estradiol (E2)-imprinted HEMA-based nanoparticles onto bacterial cellulose nanofibers (E2-NP/BC-NFs) to use for the removal of E2 from wastewater. FT-IR and NMR confirmed the structure of functional monomer. The composite system was characterised by BET, SEM, µCT, contact angle, and swelling tests. Also, the non-imprinted bacterial cellulose nanofibers (NIP/BC-NFs) were prepared to compare the results of E2-NP/BC-NFs. Adsorption of E2 from aqueous solutions was performed in batch mode and investigated via several parameters for optimization conditions. The effect of pH studies was examined in the 4.0–8.0 using acetate and phosphate buffers and a concentration of E2 of 0.5 mg/mL. The maximum E2 adsorption amount was 254 µg/g phosphate buffer at 45°C. The experimental data show that the Langmuir is a relevant isotherm model for E2 adsorption. Also, the relevant kinetic model was the pseu-do-second-order kinetic model. It was observed that the adsorption process reached equilibrium in less than 20 minutes. The E2 adsorption decreased with the increase in salt at varying salt con-centrations. The selectivity studies were performed using cholesterol and stigmasterol as com-peting steroids. The results show that E2 is 46.0 times more selective than cholesterol and 21.0 times more selective than stigmasterol. According to the results, the relative selectivity coefficients for E2/cholesterol and E2/stigmasterol were 8.38 and 86.6 times greater for E2-NP/BC-NFs than for E2-NP/BC-NFs, respectively. The synthesized composite systems were repeated ten times to assess the reusability of E2-NP/BC-NFs.

  1. The introduction could have been more attractive in term of stating the new finding and stated while this quite complicated material would be more interesting than other known effective methods.

The introduction part was revised as given below:

Molecular imprinting is a technology that creates specific recognition sites specific to the target molecule due to covalent or non-covalent interactions between the target molecule and the template molecule in the cross-linked polymer [32]. Molecular imprinted polymers (MIPs), frequently studied in recent years, are used in many application areas where high selectivity bonding is essential. They contain specific recognition sites for the molecule of interest [33,34]. BC-NFs are a 3-dimensional reticulated porous biomaterial formed by randomly arrayed nanofibers. Since BC-NFs can be considered biomaterials, interesting studies are carried out for biomedical, chromatographic, and biotechnological purposes; in addition, BC-NFs have some advantages related to cellulose fibers such as ease of manufacture and application, superior physical and chemical properties, low cost, accessible and high purity material, and increased water holding capacity due to the large number of -OH groups. Therefore, these composite structures are also preferred in adsorption studies of biomolecules due to their high specific surface area. In addition, Molecularly imprinted BC-NFs have a significant amount of application areas such as bio-separation of biological molecules, paper, filler, removal of heavy metal ions, drug delivery, packaging material, and biosensors due to their excellent surface area, high hydrophilicity, high purity, and significant chemical and mechanical stability [35–38].

Because of high selectivity, exhibited excellent template molecules adsorption capacity and fast binding kinetics, a new generation composite polymeric was developed to remove E2 for this study selectively. In this study, E2-imprinted HEMA-based nanoparticles onto bacterial cellulose nanofibers (E2-NP/BC-NFs) composite system was prepared to remove E2 from wastewater. Structure of functional monomer was confirmed by FT-IR and NMR. First, the composite system was characterized by BET, SEM, µCT, contact angle, and swelling tests. Next, the binding capacity of E2 to E2-NP/BC-NFs was investigated by selective adsorption experiments. Experiments were repeated 10 times to determine the reusability of E2-NP/BC-NFs.

  1. The preparation of 17β-Estradiol-Imprinted poly(HEMA-MATrp) nanoparticles (E2-NP) is quite complicated. Can the authors elaborate how their material reproductivity is reliable? Are there specific toolto confirm it? It should be mentioned in the methodology section.

The measurements were repeated 3 times, and the results were examined. The average of the three measurements was taken.

  1. Chemical vendor & chemical analyses are needed. It may not need to be in the manuscript, but it could have been in the supporting information. How did the authors actually measure E2 concentration with “its absorbance at 280 nm”? I assume the author used SPE for the E2 measurement? Please explain.

Thank you for asking. The experimental section was revised related to this part:

The chromatographic detection of E2, cholesterol and stigmasterol was performed using a mobile phase containing water/methanol/acetonitrile/ (60/30/10, v/v/v), the linear gradient at 0.5 mL/min flow rate by HPLC system ((Ultimate-3000, Dionex, USA). The column temperature was set at 25˚C, and the monitoring wavelength at 280 nm.

  1. What is the purpose of the experiment in Section 2.8. If the author need to prove the material reusability, why does the author need to change the contaminant and why does the author need to use the sonicator? Wouldn’t it interfere the interpretation of the adsorption studies? Please explain.

The contaminants were used for the selectivity experiment. For reusability and reproducibility studies, section 2.8 was revised, and section 2.9 was added.

2.8. Selectivity

Adsorption studies were also conducted for cholesterol (MW: 386 g/mol) and stigmasterol (MW: 412.7 g/mol) to determine the selectivity of E2-NP/BC-NFs. The chromato-graphic detection of E2, cholesterol and stigmasterol was performed using a mobile phase containing water/methanol/acetonitrile/ (60/30/10, v/v/v), the linear gradient at 0.5 mL/min flow rate by HPLC system (Ultimate-3000, Dionex, USA). The column temperature was set at 25˚C, and the monitoring wavelength at 280 nm.

2.9. Reusability and reproducibility studies

Desorption of E2 was studied with chloroform. The E2-NP/BC-NFs were immersed in a desorption medium and stirred at room temperature for 60 minutes. The final concen-tration of E2 in the desorption medium was determined using HPLC. The desorption ratio was calculated from the amount of E2 adsorbed on the particles and the final E2 concen-tration in the desorption medium. The adsorption-desorption cycle was repeated ten times for prepared E2-NP/BC-NFs to determine their reusability. After desorption, E2-NP/BC-NFs were washed with 50 mM NaOH solution to regenerate and sterilise..

  1. What is the main purpose of determining the contact angle? The authors should have given more explanation of the contact angle. It shouldn’t be for confirming the material hydrophilicity, should it?

Thank you for your comments. The relevant section was revised aas given below.

ɵ c is the contact angle, ɣ_GS is the gas-liquid surface tension, ɣ_LS is the liquid-solid surface tension, and ɣ_GL is the gas-liquid surface tension. Surfaces are described as high-energy or low-energy, and water forms a thin film on high-energy surfaces. In this case, the contact angle is zero. On low-energy surfaces, the contact angle is more significant than 90º, and the surface is hydrophobic.  The contact angle measurements for BC-NFs, E2-NP/BC-NFs, and NIP-BC-NFs composite nanofibers are 34.2°±1.2, 45.2°±2.4 and 42.7°±1.7, respectively (Figure 3. A-C). BC-NFs are highly hydrophilic due to their hydroxyl groups. As explained above, it has high energy due to its hydrophilic properties, and as a result, the contact angle value (34.2°±1.2) was found to be relatively low. It is observed that there is an increase in the contact angle value depending on adding chemicals with hydrophobic character to the structure. The angle (42.7°±1.7) increased due to the absence of E2, the template molecule in the NIP BC-NFs structure. Due to the high hydrophobic character of the E2 molecule in the E2-NP/BC-NFs structure, an increase in the contact angle (45.2°±2.4) was observed. So, it is possible to say that the structure is hydrophilic. In addition, it was understood that the increasing contact angle value shows the hydrophobic E2 molecule entered into the synthesized MIP material.

  1. E2 is known to have a very low water solubility. The detected amount is normally in a nanogram or microgram level. Please elaborate how this would affect this newly synthesized material? Can the contact angle help us explaining anything?

Thanks for your valuable comments.

The newly created substance's contact angle can reveal information about its surface characteristics and how it interacts with water. In this study, contact angle measurements used to evaluate the effectiveness of surface modification and the increase in the contact angle with E2-NPs show the existence of hydrophobic sites found in the molecularly imprinted cavities.

The following explanations were added in the conclusion section:

E2 has a low water solubility, making it difficult to remove from aqueous solutions like wastewater. The newly synthesized substance's capacity to efficiently catch and remove the hormone will depend on its solubility characteristics and how it interacts with the aqueous solution when used to remove E2 from wastewater. Due to E2's limited solubility, it may have a tendency to adsorb to solid objects or wastewater particles rather than staying in the solution. This can make it challenging to remove using standard wastewater treatment techniques. The newly created substance may be used to boost the adsorption of E2 onto its surface, enhancing the removal of the EDC from wastewater.

  1. There was insufficient discussion for the adsorption studies, especially effect of pH. Please provide more discussion.  

Thanks for your valuable comments. The following explanations were added in section 3.3.2.

“The highest E2 adsorption, as shown by the adsorption studies given in Fig 5.A , took place at pH: 7.0. which shows that interactions between E2-imprinted MIP NFs are primarily based on hydrogen bonds. The functional groups of HEMA, MATrp monomers, and E2 will not form hydrogen bonds because of the deprotonation impact at high pH. Higher or lower pH values thus impact the ligand's affinity for binding to the template molecule, reducing the selectivity of the adsorbent.”

  1. The effect of ionic strength was different between MIP and pHEMA. Why is that? Please explain.

Thank you very much for your comments. The following sentence was added in section 3.3.3.

Due to weak interactions (van der Waals interaction and hydrogen bonding) between E2 and hydroxyl groups on the surface of pHEMA nanospheres, E2 adsorption is negligible. This result also shows that no observable effect was seen on pHEMA due to the lower selectivity of E2 towards pHEMA. 

  1. Please discuss and compare (with other findings) more on the maximum adsorption capacity of the material.

Thank you very much for your comments. Section 3.3.2 was revised as given below.

Adsorption tests using various aqueous solutions of E2 at pH 7.0 and concentrations between 0.25-2.0 mg/mL were carried out to investigate the effect of E2 concentration on adsorption capacity. According to adsorption experiments using MIP NFs, E2 displays affinity binding as the number of molecules interacting with the imprinted regions increases. By filling in the specific gaps, it came to equilibrium at a concentration of 1.50 mg/mL E2. The highest E2 adsorption, as shown by the adsorption studies in Fig 5.A , took place at pH: 7.0. which shows that interactions between E2-imprinted MIP NFs are primarily based on hydrogen bonds. The functional groups of HEMA, MATrp monomers, and E2 will not form hydrogen bonds because of the deprotonation impact at high pH. Higher or lower pH values thus impact the ligand's affinity for binding to the template molecule, reducing the selectivity of the adsorbent.

Adsorption investigations were performed in the temperature range of 4-45°C to determine the effects of temperature on the adsorption capacity (Figure 5.B.). An increase in E2 adsorption was observed with increasing temperature. These findings demonstrate that incorporating E2 imprinted nanoparticles into the structure of the MATrp functional monomer results in a hydrophobic interaction between E2 and E2-NP/BC-NFs. The increased temperature enhances the adsorption capacity and accelerates the interaction kinetics between the analyte and the binding sites in hydrophobic interactions, which occur with an increase in entropy. Due to the low kinetic energy of the molecules, the hydrophobic interactions between nonpolar adsorbates and hydrophobic adsorbents may be weaker at low temperatures. As the temperature rises, the molecules' kinetic energy also rises, which may increase the hydrophobic interactions between the adsorbent and the adsorbate. As a result, the system's maximum adsorption capacity increases.

  1. I am skeptical to accept that the BET slight difference was from “the cavity formed by the imprinting of the estradiol molecule”. Please consider rewrite.

The part of the BET analysis results for this study was updated as given below:

The BET method calculated the specific surface area of E2-NP/BC-NFs. The surface morphology of BC-NFs is more oval and smooth. The arrangement of nanofibers is random to form interconnected pore structures and three-dimensional network structures with high porosity. Thus, the NF surface can become rougher. A rougher surface causes E2-NP/BC-NFs to generate larger specific surface areas than BC-NFs. This idea follows our study. The surface area of BC-NFs, NIP/BC-NFs, and E2-NP/BC-NFs was found to be 301.5, 305.2, and 313.2 m2/g, respectively, in this study [36,40,41]. In addition, the particulate systems brought into the form give the BC nanofibers the functionality to be used in different application areas. BC-NFs reveal a high surface area showing excellent potential for separation processes. With larger specific surface areas for E2-NP/BC-NFs, the biological molecules desired to be separated reach the nanofiber surface faster with rapid mass transfer. At the same time, higher adsorption capacity can be achieved. Some studies in the literature also support this idea [42,44,45]. Thus, BC-NFs reveal a high surface area showing excellent potential for separation processes in different fields of biotechnology [46,47].

  1. The NMR result has no discussion. Please provide more.

FT-IR and NMR confirmed the structure of functional monomers. NMR comments were revised as given below:

The characteristic peaks of the MATrp monomer at 1H-NMR spectra are as follows; (1) 8.22 (1H, s, N-H), (2) 7.54-7.09 (4H aromatics), (3) 6.98 (1H, d, amide NH J=5.58), (4) 5.64 (1H, t, CH2), (5) 5.32 (1H, t, CH2), (6) 4.99 (1H, m, CH) (7) 3.38(2H, dd, CH2), (8) 6.34 (1H, d, 5-ring, J=7.6), (9) 3.71 (3H, s, OCH3), (10) 1.24 (3H, t, CH3), (400 MHz, DMSO-d6). A broad singlet peak at 8.22 ppm corresponds to the NH proton of the amide group. A set of four aromatic proton signals appears as a multiplet between 7.54 and 7.09 ppm. An NH proton signal of the amide group appears as a doublet at 6.98 ppm with a coupling constant (J) of 5.58 Hz. Two methylene proton signals appear as overlapping triplets at 5.64 and 5.32 ppm. A methine proton signal appears as a broad multiplet at 4.99 ppm. Two methylene proton signals appear as a doublet of doublets at 3.38 ppm. A proton signal appears as a doublet at 6.34 ppm with a coupling constant (J) of 7.6 Hz, indicating its location in a five-membered ring. A methyl group signal appears as a singlet at 3.71 ppm. A methyl group signal appears as a triplet at 1.24 ppm. Overall, the NMR spectrum indicates that the MATrp monomer contains an amide group, an aromatic ring, and various other functional groups. The peaks are well-resolved and show distinct coupling patterns, which suggests that the compound is fairly pure and structurally well-defined.

  1. There are several inconsistency in the use of abbreviation, which makes it quite difficult to follow. Please consider reorganize it.

Thank you for your comments. All the abbreviations were reorganised.

  1. At the end, however, I'd like the authors to reflect on the value of manuscript by telling readers how all this research material could be used in practice and applied to the conventional industrial wastewater treatment plant. Without some application, the value of this work will be lost.

Thank you very much for your comments. A comparison of E2-imprinted adsorbent parts was revised as given below.

In the literature, studies are reported on the selective removal of EDCs. As different adsorbents have been proposed for the adsorption of E2, the importance of studies using molecularly imprinted polymers is emerging. Farber et al. prepared MIPs for 17β-estradiol removal using covalent and non-covalent interactions. For this purpose, 4-vinyl benzene as a covalent monomer, besides methacrylic acid as a non-covalent monomer, ethylene glycol dimethacrylate crosslinker, and acetonitrile were used as porogen. It was concluded that the polymers obtained have the highest adsorption capacity (10.73 μg/g) due to the imprinting process using covalent interactions [44]. Denizli et al., the obtained polymers have the highest adsorption capacity (10.73 μg/g) due to the covalent interactions' imprinting process. E2 imprinted particle-embedded poly(HEMA) cryogels prepared for 17β-Estradiol (E2) removal. The composite cryogels achieved an adsorption capacity of 5.32 mg/g. As a result of the adsorption studies performed in the presence of cholesterol and stigmasterol selected as competitive molecules, it was concluded that there is 7.6 times more selective binding to cholesterol and 85.8 times more to stigmasterol.[45]. In the study using lake water as a natural source, Meng et al. prepared estradiol-imprinted microspheres using the non-covalent imprinting technique. The maximum adsorption capacity for estradiol reached 380 nmol/g using MIP microspheres. Furthermore the imprinting factor was calculated as 35 [46]. Lai et al., E2 imprinted methacrylate-based microparticles were prepared by Lai et al.. As a result of adsorption studies, it was reported that high recovery rates of up to 97% were obtained, and the adsorption capacity was determined as 15 mg/g [47]. Xiong et al. reported that switchable zipper-like thermoresponsive molecularly imprinted polymers reach higher adsorption capacity (8.78 mg/g) and more substantial selectivity (imprinting factor was 3.18) [48]. Gao et al. prepared magnetic MIPs using nanoparticles of Fe3O4 with acrylamide as a functional monomer to separate and determine E2 from milk. It was reported that the adsorption performance of the MIP system was 12.62 mg/g [49]. He et al. reported using core-shell molecularly imprinted polymers fixed to the SiO2 surface to separate and purify β-estradiol from marine sediment for the first time. Studies using 3-aminopropyltrimetyoxysilane as a coupling agent with methacrylic acid as a functional monomer were performed with MIP solid-phase extraction method with HPLC.  This study's result determined that the adsorption capacity for the MIP system was 468.3 µg/g, while it was 146.7 µg/g for NIP [50]. Xia et al. used surface-modified magnetic nanoparticles as core-shell MIPs by ultrasonication-assisted synthesis to selectively remove E2 from aqueous media. The results show that  E2-imprinted MIPs achieved a good recovery of 95.8% [51]. Peng et al. prepared photonic-magnetic responsive molecularly imprinted microspheres via seed polymerization to the removal of E2. It was reported that MIP-microspheres show good adsorption characteristics for E2 with an adsorption capacity of 0.84 mg/g) with fast binding kinetics (Kd = 26.08 mg L−1) [52]. In a study using MIP created by surface imprinting of magnetic graphene oxide, it was reported that E2 was removed quickly and selectively from acetonitrile solutions of E2. Adsorption studies show that the kinetic value and binding capacity of MIP system were 0.0062 g (mg/min) and 4.378 μmol/g, respectively [53]. Our study achieved the maximum E2 adsorption amount of 254 µg/g at 45°C using E2-Imprinted Nano-Particles onto Bacterial Cellulose Nanofibers. So, these results show that E2-NP/BC-NFs have very promising adsorbents showing comparable performance in the selective and effective removal of E2 as useful adsorbents, which is highly attractive in the literature.

Table 4. Comparison of removal capacity of 17β-Estradiol with different E2-imprinted adsorbents.

Method/Functional monomer/Polymer

% Removal

Sample

Ref

Covalent MIP/ 4-vinyl benzene-methacrylic acid /polymer

10.73 μg/mg

Aqueous media

[44]

MIP particle-embedded poly(HEMA) cryogel

5.32 mg/g

Aqueous media

[45]

Non covalent MIP/ Acrylamide- trimethylpropanol trimethacrylate/microsphere

380 nmol/mg

Aqueous media

[46]

MIP/ methacrylic acid-ethylene glycol dimethcarylate/ submicron particles

15 mg/g

Aqueous media

[47]

stimuli-responsive MIP/Acrylamide-2- 2-acrylamide-2-methyl propane sulfonic acid/polymer

8.78 mg/g

Acetonitrile solutions

[48]

Surface MIP/ Fe3O4@Acrylamide/nanoparticle

12.62 mg/g

Milk

[49]

Core-shell MIP/ 3-aminopropyltrimetyoxysilane-methacrylic acid/polymer

468.3 µg/g

Marine sediment

[50]

Core-shell MIP/ methacrylic acid-ethylene glycol dimethcarylate /magnetic nanoparticle

>95

Aqueous media

[51]

Photonic-magnetic responsive MIP/ Fe3O4@SiO2-KH570-4-[(4-methacryloyloxy) phenylazo]benzoic

acid /nanoparticle

315.6 ± 10.1 μg/g

Milk

[52]

Nanosized substrate MIP /Acrylamide- -GO-Fe3O4 /nanoparticle

4.378 µmol/g

Acetonitrile solutions

[53]

Nanoparticle MIP/HEMA-MATrp/bacterial cellulose nanofiber

254 µg/g

Aqueous media

Our study

Round 2

Reviewer 2 Report

I am satisfied with the latest version that the authors provided. Therefore, I accept it as is.